# Domain Specific Question Answering Over Knowledge Graphs Using Logical Programming and Large Language Models

**Navid Madani, Kenneth Joseph, Rohini K. Srihari**

Computer Science and Engineering
Davis Hall, Buffalo, New York 14260-2500
{smadani,kjoseph,rohini}@buffalo.edu

## Abstract

Answering questions over domain-specific graphs requires a tailored approach due to the limited number of relations and the specific nature of the domain. Our approach integrates classic logical programming languages into large language models (LLMs), enabling the utilization of logical reasoning capabilities to tackle the KGQA task. By representing the questions as Prolog queries, which are readable and near close to natural language in representation, we facilitate the generation of programmatically derived answers. To validate the effectiveness of our approach, we evaluate it using a well-known benchmark dataset, MetaQA. Our experimental results demonstrate that our method achieves accurate identification of correct answer entities for all test questions, given only a very small fraction of the training data. Overall, our work presents a promising approach to addressing question answering over domain-specific graphs, offering an explainable and robust solution by incorporating logical programming languages. Code and models are publicly available on Github.[1]

## Introduction

Question Answering over Knowledge Graphs (KGQA) poses significant challenges in the field of Natural Language Processing (NLP). As structured knowledge graphs capturing rich semantic information become prevalent, there is a pressing need for intelligent systems that can reason effectively and provide accurate answers to intricate questions within specific domains. The primary focus of KGQA is to bridge the gap between human language and structured knowledge representations. When presented with a question in natural language, KGQA systems aim to traverse the knowledge graph consisting of entities and their relationships, extracting relevant information to generate precise answers. This task demands not only language comprehension but also the ability to perform logical reasoning across the edges of the graph to derive meaningful insights. Although large language models (LLMs) powered by deep learning have shown remarkable capabilities in natural language understanding and generation, they may not be specifically trained on a particular knowledge source or possess a deep understanding of domain-specific facts. However, LLMs can serve as a valuable tool to represent questions within a domain, extracting query or question meanings. Leveraging logical programming approaches, these representations can be processed to handle reasoning and knowledge representation. One notable advantage of this approach is that it empowers users of the system to manage knowledge dynamically. They can modify, delete, or add new entries into the knowledge graph without requiring changes to the system itself. By integrating logical programming techniques with LLMs, KGQA systems gain the flexibility to adapt to evolving knowledge requirements while maintaining their functionality.

In this paper, we address the challenges of domain-specific KGQA by combining the strengths of large language models and logical programming. We propose an approach that utilizes LLMs to represent questions within a specific domain, extracting their meanings, while employing logical programming techniques for reasoning and knowledge representation. Our objective is to demonstrate how this integration enables robust and adaptable KGQA systems that can navigate domain-specific knowledge graphs and provide accurate answers to complex questions. To evaluate the effectiveness of our proposed approach, we conduct experiments using the MetaQA dataset (Zhang et al. 2018), a widely adopted benchmark in KGQA research. By comparing our method against state-of-the-art approaches, we demonstrate its capability to accurately identify the correct answer entities for a range of questions. Notably, our experiments show promising results even when our model is trained on only a small fraction of the available training data, indicating its efficiency and generalization ability.

The contributions of this paper are two-fold:

- We propose a novel approach to equip LLMs with logical programming languages for domain-specific KGQA, enhancing their reasoning capabilities and providing explainable solutions.

- We demonstrate the effectiveness of our approach through comprehensive experiments on the MetaQA dataset, showcasing its ability to accurately represent questions given only a small set of annotated data.

[1]https://github.com/navidmdn/logic_based_qa

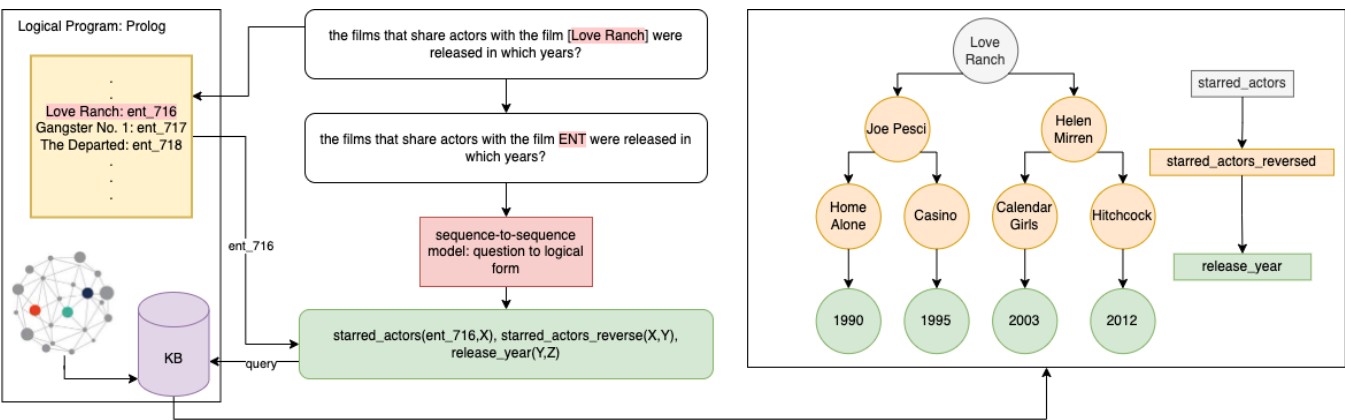

Figure 1: The complete inference pipeline of our proposed method. Note that the inference tree on the right side is a subset of the answer drawn here to clarify the schema of the model's output.

## Related Work

A variety of approaches have been taken to address the problem of multi-hop question answering. A number of prior works have used graph embedding models to encode entities and relations in a knowledge graph and then score the triples in a KG and construct a scoring function so that the score for a correct triple is higher than the score of an incorrect one (Nickel, Tresp, and Kriegel 2011; Yang et al. 2014a; Balazevic, Allen, and Hospedales 2019; Dettmers et al. 2017; Vashishth et al. 2019). Others have approached the problem by constructing a function that maps the question embedding along with an embedding of the graph or a subgraph around the question entity to the answer entity's embedding in knowledge graph (Sun et al. 2018; Saxena, Tripathi, and Talukdar 2020; Sun, Bedrax-Weiss, and Cohen 2019; He et al. 2021). Still others adapt a slightly different method, training a teacher model to learn intermediate signals and a student model to answer the questions (He et al. 2021). There also has been efforts by (Xie, Hao, and Zhang 2022) that pushed the performance of these models on 2 and 3 hop splits to the limits. They propose a sequential reasoning self-attention mechanism which is guided by a GRU-inspired Flow Control (GFC) and their work is inspired by (Shi et al. 2021). Finally, most relevant to our work, (Yang et al. 2014b) and (Yang et al. 2015) try to learn the logical form of the natural language questions by building a semantic embedding space. However, our work differs from theirs in that we use LLMs to represent the question in logical form instead of manually building a semantic mapping space. The present work is thus the first to use large language models to represent questions in logical form and equip LLMs with logical programming tools to answer questions.

## Dataset

The MetaQA dataset is a widely used benchmark dataset for question answering over knowledge graphs (KGQA). The MetaQA dataset consists of questions that require reasoning over a given knowledge graph. The knowledge graph represents a structured database of 134,741 facts and 9 relations,

| MetaQA | train | dev | test |
|--------|-------|-----|------|
| 1-hop | 96106 | 9992 | 9947 |
| 2-hop | 118980 | 14872 | 14872 |
| 3-hop | 114196 | 14274 | 14274 |

Table 1: Statistics for MetaQA dataset

providing a rich source of information for answering domain specific questions. Each question in the MetaQA dataset is associated with the provided knowledge graph, comprising entities, relations, and their connections. The dataset incorporates a diverse range of questions, covering various domains and types of queries. These questions often involve multiple hops or intermediate steps to reach the correct answer. These multi-hop paths guide the reasoning process required to answer the questions accurately. By traversing these paths, the model must navigate through different entities and relations to arrive at the correct answer. The dataset also provides the intermediate steps that leads us from question entity to the answer . This is one of the important reasons that we chose MetaQA dataset. Table 1 briefly describes the statistics of this dataset.

## Approach

### Question to Logical Form Annotation

Each question in MetaQA dataset comes with the inference path inside the knowledge graph. For example, for the 2-hop question *"the movies written by [Hilary Brougher] were directed by who?"* there exists an inference path of *writer_movie_director* which shows the sequence of relations we need to traverse in the graph to reach the answer entity from the question entity *Hilary Brougher*. We use this inference path and annotate the question with the correct prolog query. To do so, we first break down the inference path into pairs. For the example above we would get *writer_movie* and *movie_director* pairs. Then we map each of the pairs to their corresponding predicate. Table 2 provides a list of all mappings that are available in the dataset. To

ensure that the model focuses solely on the representation of the question itself, we employ a substitution strategy. Specifically, we replace the question entity with a designated string placeholder denoted as *ENT*. For example, if we have the question "Which movies directed by [ENT] were written by whom?" we construct the corresponding Prolog query as **directed_by_reverse(ENT, X), written_by(X, Y)**. This query captures the essence of the original question while preserving its logical structure.

| Inference Pair | Predicate |
|---|---|
| actor_movie | starred_actors_reverse |
| director_movie | directed_by_reverse |
| movie_actor | starred_actors |
| movie_director | directed_by |
| movie_genre | has_genre |
| movie_imdbrating | has_imdb_rating |
| movie_imdbvotes | has_imdb_votes |
| movie_language | in_language |
| movie_tags | has_tags |
| movie_writer | written_by |
| movie_year | release_year |
| tag_movie | has_tags_reverse |
| writer_movie | written_by_reverse |

Table 2: Mapping between different inference pairs and Prolog predicates

## Question to Logical Form Translation

To facilitate the translation of questions into their corresponding logical forms, we begin by developing a question comprehension module. To accomplish this task, we harness the power of encoder-decoder transformer models, known for their exceptional potential in sequence to sequence transformation ability (Vaswani et al. 2017). To collect the dataset necessary for fine-tuning a sequence-to-sequence transformer model, we leverage the multi-hop path information provided by the MetaQA dataset and annotate each question with the corresponding query as described in the previous section. From the total pool of 329,282 multi-hop training examples in MetaQA, we randomly sample and annotate subsets consisting of 100, 250, 500, and 1000 samples with each subset consisting of equal number of examples from each of the 1, 2 and 3 hop samples. These subsets are respectively labeled as *s100, s250, s500* and *s1000*. To transform the question into an intermediate query representation, we employ a T5-small sequence-to-sequence transformer model (Raffel et al. 2019). This model effectively learns to generate accurate representations of the questions, serving as a bridge between natural language input and logical query output. For the training process, we fine-tune the model using each of the annotated training sets, iterating through 5000 training steps. The best-performing model is selected based on the exact match score obtained from the development dataset.To optimize the model's performance, we utilize the AdamW optimizer with an initial learning rate of 5e-5. Additionally, a linear learning rate scheduler is em-

ployed. The training is conducted using a batch size of 8, making efficient use of a single A100 GPU for computational acceleration.

## Question Answering

The question answering process in our proposed model is illustrated in Figure 1. To begin, we transform each triple in the knowledge base of the MetaQA dataset into a first-order logic predicate. For instance, given the triple *(Innocence — written_by — Hilary Brougher)*, we construct the corresponding predicate **written_by(Innocence, Hilary Brougher)**.

When processing a specific question, we generate its logical form using the transformer model. The logical form provides a structured representation of the question's meaning. Subsequently, we replace the *ENT* token in the logical form with the corresponding entity ID from the knowledge graph. This substitution results in the final Prolog query. Finally, we execute the Prolog query, which involves querying the knowledge graph. By executing the query, we retrieve both the answers to the question and the logical path that connects the question entity to the answer entities. This path provides valuable insights into the reasoning process and the information flow within the knowledge graph.

## Experiments and Results

MetaQA questions mostly come with multiple answers. Prior methods have used hit@1 as a metric to measure the performance of their model. This means that they measure if the highest ranked entity given by their model exists in the answer set. Our approach produces the exact solution path inside the knowledge graph and consequently it outputs all of the answers to the question instead of producing a score distribution over graph entities (as depicted in Figure 1). For the sake of comparison, we also measure the hit@1 metric for our model over multi hop test datasets. In other words, we randomly pick one of the answer entities and assume it is the rank 1 answer of the model and consequently we calculate the hit@1 score. Table 3 compares our method with prior work.

In order to get robust results we repeat the process of sampling training data and annotating it 5 times and each time we sampled 100, 250, 500 and 1000 samples. The sampling process was straightforward; each time we sample randomly and equally from each of the 1-hop, 2-hop and 3-hop training datasets. We also annotated 3000 samples from the validation and test set of the MetaQA dataset. Figure 2 shows the variance of performance on each of these datasets. Since the variance was high on the test set with 100 samples we only reported s250, s500 and s1000 in table 3. According to these results the 3-hop test set's representation is the easiest to learn since it doesn't come with many variations of natural language to describe. On the other hand the 2-hop dataset is the hardest to learn. However, all of these samples are collected randomly. But with a manual and careful sample collection, we can see that even 500 samples are enough to learn the whole dynamics of this dataset and learn to represent questions in logical form. We conclude that our

| Models | MetaQA-1hop | MetaQA-2hop | MetaQA-3hop |
|---|---|---|---|
| GraftNet (Sun et al. 2018) | 97.0 | 94.8 | 77.7 |
| PullNet (Sun, Bedrax-Weiss, and Cohen 2019) | 97.0 | 99.9 | 91.4 |
| EmbedKGQA (Saxena, Tripathi, and Talukdar 2020) | 97.5 | 98.8 | 94.8 |
| NSM (He et al. 2021) | 97.1 | 99.9 | 98.9 |
| TransferNet (Shi et al. 2021) | 97.5 | 100.0 | 100.0 |
| GFC (Xie, Hao, and Zhang 2022) | 97.7 | 100.0 | 100.0 |
| T5-small+prolog+250 samples | 98.67 | 97.77 | **100.0** |
| T5-small+prolog+500 samples | **100.0** | 99.33 | **100.0** |
| T5-small+prolog+1000 samples | **100.0** | **100.0** | **100.0** |

Table 3: Comparison of hit@1 score of previous methods compared to our method over multi-hop test datasets. The scores for the best model among 5 iterations of sampling is reported for our proposed method.

model is capable of correctly answering all questions in the test dataset with only 1000 annotated examples.

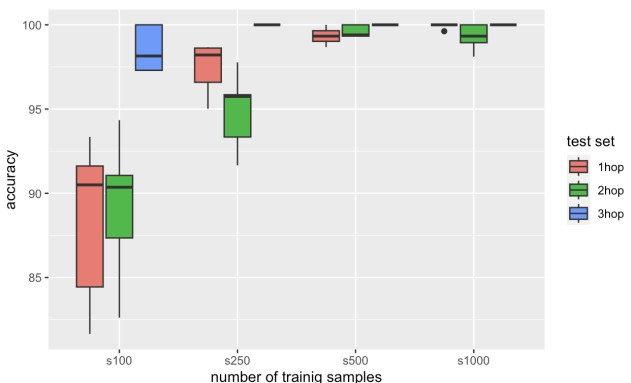

Figure 2: The variance of hit@1 of each model based on the number of training examples available in each training dataset

## Robustness of the method

In order to get robust results we repeat the process of sampling training data and annotating it 5 times and each time we sampled 100, 250, 500 and 1000 samples. The sampling process was straightforward; each time we sample randomly and equally from each of the 1-hop, 2-hop and 3-hop training datasets. We also annotated 3000 samples from the validation and test set of the MetaQA dataset. Figure 2 shows the variance of performance on each of these datasets. Since the variance was high on the test set with 100 samples we only reported s250, s500 and s1000 in table 3. According to these results the 3-hop test set's representation is the easiest to learn since it doesn't come with many variations of natural language to describe. On the other hand the 2-hop dataset is the hardest to learn.

However, all of these samples are collected randomly. But with a manual and careful sample collection, we can see that even 500 samples are enough to learn the whole dynamics of this dataset and learn to represent questions in logical form.

## Conclusion

In this work, we have presented a framework that leverages logical programming languages as a powerful tool for large language models (LLMs) for domain specific question answering over knowledge graphs. By utilizing logical programming languages such as Prolog which benefits from the inherent similarity between the representations of meaning in logical programming languages and natural language, we have showcased the ability to bridge the gap between natural language understanding and logical reasoning. We evaluated our model on a relatively small dataset and showed that it is able to fully answer questions given a small subset of annotated representations due to the pre-trained knowledge encoded even in relatively small LLMs.

## Limitations

MetaQA dataset is a synthesized dataset focused on the movie domain. Although it provides a comprehensive evaluation environment for a domain-specific question answering over knowledge graphs, it may not capture the full complexity and diversity of real-world scenarios. For instance, it does not encompass a wide range of relations found in open-domain datasets like WebQuestions, which are based on Freebase and cover a broader domain. To mitigate this limitation, future research could explore approaches such as relation and entity matching. By incorporating techniques to match entities and relations in a more flexible and adaptive manner, our model could potentially be extended to handle datasets like WebQuestions and address a broader range of real-world KGQA scenarios.

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
