# OpenReview forum: "Domain Specific Question Answering Over Knowledge Graphs Using Logical Programming and Large Language Models"
_AAAI.org/2024/Workshop/NuCLeaR — NuCLeaR 2024_

### Official Review · Reviewer_Eknt · 2023-12-07
**Question Answering Over Knowledge Graphs**

**Rating:** 6
**Confidence:** 4

**Review:**

Straight forward, early work. Could use experiments with harder datasets (as mentioned towards the end of the paper), more discussion on next steps, comparisons to other domains, etc. It's also short with more spacing than other papers suggesting more content should be added

---

### Official Review · Reviewer_16Td · 2023-12-08
**Domain Specific Question Answering Over Knowledge Graphs Using Logical Programming and Large Language Models**

**Rating:** 3
**Confidence:** 5

**Review:**

Scope:
The manuscript deals with narrow AI for answering questions over domain specific Knowledge graph. The motivation of the research is to devise a way to reduce hallucinations from LLM responses, find a Knowledge graph for question answering system (KGQA) to increase the factuality of LLMs using logical programming.

Strength:

•	The experimental results were reported on standard MetaQA dataset. The MetaQA dataset consists of questions that require reasoning over a given knowledge graph having 134,741 facts and 9 relation.
•	Reduce hallucinations from LLM responses, find a KGQA to increase the factuality of LLMs using logical programming.

Weakness:

1.	There is little novelty in the manuscript. Please justify the novelty of your work.

2.	Architecture diagram of the proposed methodology is missing. The author is suggested to draw the architecture diagram highlighting the contribution of author in existing area of research.


3.	To further validate the performance of system MFAQ dataset may be used along with MetaQA. MFAQ is a multilingual FAQ dataset publicly available. It contains around 6M FAQ pairs from the web, in 21 different languages. Although this is significantly larger than existing FAQ retrieval datasets, it comes with its own challenges: duplication of content and uneven distribution of topics.

4.	How your work is different from Mihindukulasooriya, N., Tiwari, S., Enguix, C.F. and Lata, K., 2023, October. Text2kgbench: A benchmark for ontology-driven knowledge graph generation from text. In International Semantic Web Conference (pp. 247-265). Cham: Springer Nature Switzerland.

5.	The mathematical representation of the manuscript is weak. Refine the equations and add more logical representations of your work.

6.	Author need to further elaborate their findings with evaluation measures like BLEU METEOR, ROUGE, Exact Match, accuracy (A), hallucination rate (H), and missing rate (M) for better evaluation of the proposed work.

6.	The paper is not written well, there is lack of clarity, cohesion, and connectivity. Improve the writing of paper to help the reader to understand the methodology.

7.	The related work section is  on weaker side and  author is suggested to update it with recent research paper as listed below:

I.	Mihindukulasooriya, N., Tiwari, S., Enguix, C.F. and Lata, K., 2023, October. Text2kgbench: A benchmark for ontology-driven knowledge graph generation from text. In International Semantic Web Conference (pp. 247-265). Cham: Springer Nature Switzerland.

II.	Wu, Y., Hu, N., Qi, G., Bi, S., Ren, J., Xie, A. and Song, W., 2023. Retrieve-Rewrite-Answer: A KG-to-Text Enhanced LLMs Framework for Knowledge Graph Question Answering. arXiv preprint arXiv:2309.11206.

III.	Heyi, Z.H.A.N.G., Xin, W.A.N.G., Lifan, H.A.N., Zhao, L.I., Zirui, C.H.E.N. and Zhe, C.H.E.N., 2023. Research on Question Answering System on Joint of Knowledge Graph and Large Language Models. Journal of Frontiers of Computer Science & Technology, 17(10), p.2377.

IV.	Sun, K., Xu, Y.E., Zha, H., Liu, Y. and Dong, X.L., 2023. Head-to-Tail: How Knowledgeable are Large Language Models (LLM)? AKA Will LLMs Replace Knowledge Graphs?. arXiv preprint arXiv:2308.10168.

V.	Taffa, T.A. and Usbeck, R., 2023. Leveraging LLMs in Scholarly Knowledge Graph Question Answering. arXiv preprint arXiv:2311.09841.

VI.	Lehmann, J., Gattogi, P., Bhandiwad, D., Ferré, S. and Vahdati, S., 2023, September. Language Models as Controlled Natural Language Semantic Parsers for Knowledge Graph Question Answering. In European Conference on Artificial Intelligence (ECAI) (Vol. 372, pp. 1348-1356). IOS Press.


Overall assessment:
Not recommended to be published in NuCleaR 2024.

---

### Decision · Program_Chairs · 2023-12-11

Accept